# On the Use of Concentrated Time–Frequency Representations as Input to a Deep Convolutional Neural Network: Application to Non Intrusive Load Monitoring

**DOI:** 10.3390/e22090911

**Published:** 2020-08-19

**Authors:** Sarra Houidi, Dominique Fourer, François Auger

**Affiliations:** 1Laboratoire IBISC (Informatique, BioInformatique, Systèmes Complexes), EA 4526, University Evry/Paris-Saclay, 91020 Evry CEEEE, France; Sarra.Houidi@gmail.com; 2Institut de Recherche en Energie Electrique de Nantes Atlantique (IREENA), EA 4642, University of Nantes, 44602 Saint-Nazaire, France; Francois.Auger@univ-nantes.fr

**Keywords:** non-intrusive load monitoring (NILM), time–frequency representation (TFR), deep learning, convolutional neural network (CNN), synchrosqueezing, Layer wise relevance propagation (LRP)

## Abstract

Since decades past, time–frequency (TF) analysis has demonstrated its capability to efficiently handle non-stationary multi-component signals which are ubiquitous in a large number of applications. TF analysis us allows to estimate physics-related meaningful parameters (e.g., $F_{0}$, group delay, etc.) and can provide sparse signal representations when a suitable tuning of the method parameters is used. On another hand, deep learning with Convolutional Neural Networks (CNN) is the current state-of-the-art approach for pattern recognition and allows us to automatically extract relevant signal features despite the fact that the trained models can suffer from a lack of interpretability. Hence, this paper proposes to combine together these two approaches to take benefit of their respective advantages and addresses non-intrusive load monitoring (NILM) which consists of identifying a home electrical appliance (HEA) from its measured energy consumption signal as a “toy” problem. This study investigates the role of the TF representation when synchrosqueezed or not, used as the input of a 2D CNN applied to a pattern recognition task. We also propose a solution for interpreting the information conveyed by the trained CNN through different neural architecture by establishing a link with our previously proposed “handcrafted” interpretable features thanks to the layer-wise relevant propagation (LRP) method. Our experiments on the publicly available PLAID dataset show excellent appliance recognition results (accuracy above 97%) using the suitable TF representation and allow an interpretation of the trained model.

## 1. Introduction

Real-world signals such as electrical signals can be modeled as a mixture of non-stationary components which can be disentangled using efficient methods allowing to jointly represent them in time and in frequency. Among popular time–frequency (resp. time–scale) analysis techniques, Short-time Fourier Transform (STFT) and Continuous Wavelet Transform (CWT), which are rooted in a mature mathematical background, have shown their efficiency in a large number of applications [1,2] (e.g., audio processing). However, such techniques suffer from theoretical limitations, mostly explained by the Heisenberg-Gabor uncertainty principle which can be compensated using promising post-processing methods such as reassignment and synchrosqueezing [3,4]. These methods were proposed to sharpen a Time–Frequency (TF) representation by moving the transforms values to more accurate coordinates closer to the exact TF support of the signal, resulting in an improved readability. More particularly, the success of the synchrosqueezing transform comes from its capability to compute sharpened reversible Time–Frequency Representations (TFRs), which enable advanced processing methods for disentangling and extracting the elementary signal components (modes) [5,6].

Nowadays, to the best of our knowledge, there exists no study which investigates the relevance of such post-processing techniques when they are combined with a machine learning framework based on Convolutional Neural Network (CNN) in a realistic application scenario [7]. Moreover, representation learning is an active research subject [8] because it addresses the question of the learning model explainability and interpretability. Hence, developing understandable models is full of interest since it enables the validation of machine learning algorithms and paves the way to robust and efficient methods. This study aims at filling the gap by investigating a hybrid approach which consists of combining time–frequency signal processing techniques with Deep Neural Network (DNN) architectures. We address Non Intrusive Load Monitoring (NILM) [9] as a “toy” problem, which consists of a Home Electrical Appliance (HEA) identification from the observation of voltage and current measurements. NILM that is full of interest for smart grid applications, allows a better control and understanding of the electricity consumption through a possible real-time feedback. In the literature, most of the state-of-the-art methods disaggregate the observed energy curve measured at the building electrical panel by computing signal features that allow the recognition of the HEA signature [10,11,12]. Previous works aim at computing an effective set of electrical features that are used in a machine learning framework [13,14,15]. A recent study [16] proposes a deep learning approach based on CNN applied on a so called “VI trajectory” for which the authors report an average F-measure of 77.60% on the publicly available PLAID dataset [17].

Thus, investigating HEA identification allows us to comparatively assess in a practical scenario the use of several STFT-based TF representations with different TF concentration and to establish links with our previous works [15,18,19] based on “handcrafted" physically meaningful electrical features computed from the active and reactive powers. Our Time–Frequency Representation (TFR) are investigated with several deep CNN architectures.

This paper is organized as follows. In Section 2, we recall the definitions of the STFT-based reassignment and the synchrosqueezing methods. In Section 3, we address the NILM problem, for which we propose several techniques based on different CNN architectures and different TF representations as inputs. We comparatively assess the investigated methods in Section 4. Section 4.3 describes the Layer-wise Relevance Propagation (LRP) method that is used to inspect the CNN feature selection, and the obtained results are analyzed. Finally, in Section 5, we discuss our approach and results with a consideration for future work.

## 2. STFT-Based Time–Frequency Analysis

Here, we define the tools allowing us to compute the TF representation of a signal that is considered as the input of a neural network architecture.

### 2.1. Short-Time Fourier Transform (STFT)

Given a signal *x*, we define its STFT $F_{x}^{h}\left(t,\omega \right)$ at any time *t* (expressed in seconds) and frequency ω (expressed in rad.s${}^{-1}$), using a differentiable analysis window *h*, as: (1)$$\begin{array}{cc} F_{x}^{h}\left(t,\omega \right) & =\int_{R}x\left(u\right)h{\left(t-u\right)}^{*}\;e^{-j\omega u}\;du, \end{array}$$(2)$$\begin{array}{cc} \; & =\;e^{-j\omega t}\int_{R}x\left(t+u\right)h{\left(-u\right)}^{*}\;e^{-j\omega u}\;du \end{array}$$
with $j^{2}=-1$, and $z^{*}$ being the complex conjugate of *z*. Thus, a TF representation is provided by the spectrogram that can be computed as $|F_{x}^{h}{\left(t,\omega \right)|}^{2}$.

The STFT of a signal admits the following synthesis formula when $h(t_{0})\neq 0$, allowing to recover the signal *x* with a delay $t_{0}\geq 0$:(3)$$x\left(t-t_{0}\right)=\frac{1}{h(t_{0})}\int_{-\infty }^{+\infty }F_{x}^{h}\left(t,\omega \right)\;e^{j\omega (t-t_{0})}\;\frac{d\omega }{2\pi },$$
which remains valid for any differentiable analysis window *h*. In practice, $t_{0}=0$ is commonly used for any symmetrical analysis window but, we show in [20] that a suitable choice is $t_{0}=\arg\max_{t}h\left(t\right)$, that can be obtained by solving $\frac{dh}{dt}\left(t\right)=0$. In this study, we use the Gabor transform that is defined for a Gaussian analysis window expressed as:(4)$$h\left(t\right)=\frac{1}{\sqrt{2\pi }T}\;e^{-\frac{t^{2}}{2T^{2}}},$$
where parameter *T* corresponds to the time-spread of the window.

### 2.2. Reassignment and Synchrosqueezing

Reassignment and synchrosqueezing are two sharpening techniques designed to improve the readability of a STFT [4]. The main difference between these methods is the reconstruction capability of the synchrosqueezing in comparison to the reassignment method which provides non reversible TF representations. The reversibility property can be satisfied by preserving the phase information by moving the transform instead of its squared modulus, and through a trade-off which consists of moving the values along a unique axis direction (frequency). As a result, the synchrosqueezing transform has a slightly poorer concentration in the time–frequency plane (depending on the waveform of the analyzed signal) in comparison to the reassignment method, however, it admits a signal reconstruction formula.

### 2.3. Reassignment

The STFT-based reassignment method consists in moving the values of the spectrogram according to the map $\left(t,\omega \right)\mapsto \left(\hat{t}\left(t,\omega \right),\hat{\omega }\left(t,\omega \right)\right)$, where $\hat{t}\left(t,\omega \right)=ℜ\left(\tilde{t}\left(t,\omega \right)\right)$ and $\hat{\omega }\left(t,\omega \right)=ℑ\left(\tilde{\omega }\left(t,\omega \right)\right)$, are the so-called reassignment operators. The reassignment operators correspond to instantaneous frequency and group-delay estimators computed at any TF point as [21]: (5)$$\begin{array}{cc} \tilde{t}\left(t,\omega \right) & =t-\frac{F_{x}^{Th}\left(t,\omega \right)}{F_{x}^{h}\left(t,\omega \right)}, \end{array}$$(6)$$\begin{array}{cc} \tilde{\omega }\left(t,\omega \right) & =j\omega +\frac{F_{x}^{Dh}\left(t,\omega \right)}{F_{x}^{h}\left(t,\omega \right)}, \end{array}$$
with Th(t)=th(t) and $Dh\left(t\right)=\frac{dh}{dt}\left(t\right)$. Thus, the reassigned spectrogram is obtained by computing:(7)$${\mathrm{RF}}_{x}\left(t,\omega \right)=\int \;\;\;\;\;\int_{R^{2}}{\left|F_{x}^{h}\left(\tau ,\Omega \right)\right|}^{2}\delta \left(t-\hat{t}\left(\tau ,\Omega \right)\right)\delta \left(\omega -\hat{\omega }\left(\tau ,\Omega \right)\right)d\tau d\Omega .$$

### 2.4. Synchrosqueezing

The synchrosqueezing transform of the STFT can be deduced from the simplified synthesis formula given by Equation (Equation 3). Hence, for any $t_{0}\geq 0$ such that $h(t_{0})\neq 0$, the synchrosqueezed STFT can be defined as [22]:(8)$${\mathrm{SF}}_{x}^{h}\left(t,\omega \right)=\int_{R}F_{x}^{h}\left(t,\omega^{\prime }\right)\;e^{j\omega^{\prime }\left(t-t_{0}\right)}\delta \left(\omega -\hat{\omega }\left(t,\omega^{\prime }\right)\right)\;d\omega^{\prime }.$$

As a result, we can obtain a sharpened TF representation by computing $|{\mathrm{SF}}_{x}^{h}{\left(t,\omega \right)|}^{2}$. The most important property of ${\mathrm{SF}}_{x}^{h}\left(t,\omega \right)$ is its reconstruction capability which leads to:(9)$$\hat{x}\left(t-t_{0}\right)=\frac{1}{h{\left(t_{0}\right)}^{*}}\int_{R}{\mathrm{SF}}_{x}^{h}\left(t,\omega \right)\frac{d\omega }{2\pi }$$
and allows us to recover the whole signal *x*. The signal components can also be individually extracted as proposed in [23] when the integration area of Equation (Equation 9) is restricted to the vicinity of each ridge.

### 2.5. Time-Reassigned Synchrosqueezing

In 2019, He et al. proposed, in [24], a new variant of the synchrosqueezing technique which overcomes the problem of non reversibility by moving the transform values along the time axis instead of the frequency axis. Hence, the time-reassigned synchrosqueezed STFT (also called first-order horizontal synchrosqueezing) can be defined as [25]:(10)$$S_{x}^{h}\left(t,\omega \right)=\int_{R}F_{x}^{h}\left(\tau ,\omega \right)\delta \left(t-{\hat{t}}_{x}\left(\tau ,\omega \right)\right)\;d\tau$$
where ${\hat{t}}_{x}\left(t,\omega \right)$ corresponds to the time reassignment operator which is classically computed using Equation (Equation 5). The marginalization over time of the resulting transform leads to:(11)$$\begin{array}{cc} \int_{R}S_{x}^{h}\left(t,\omega \right)dt & =\int \;\;\;\;\;\int_{R^{2}}F_{x}^{h}\left(\tau ,\omega \right)\delta \left(t-{\hat{t}}_{x}\left(\tau ,\omega \right)\right)\;dtd\tau \end{array}$$(12)$$\begin{array}{cc} \; & =\int_{R}F_{x}^{h}\left(\tau ,\omega \right)d\tau =F_{h}{\left(0\right)}^{*}F_{x}\left(\omega \right). \end{array}$$

Hence, an exact signal reconstruction can be obtained using the inverse Fourier transform as [25]:(13)$$x\left(t\right)=\frac{1}{2\pi F_{h}{\left(0\right)}^{*}}\int \;\;\;\;\;\int_{R^{2}}S_{x}^{h}\left(\tau ,\omega \right)\;e^{j\omega t}\;d\tau d\omega .$$

### 2.6. Discretization

In real-world application scenarios such as NILM, continuous signals are discretized according to a given sampling rate $F_{s}$ (expressed in Hz). Thus, all the previously defined expressions can be reworded using the rectangle approximation method as $F_{x}^{h}\left[n,m\right]\approx F_{x}^{h}\left(\frac{k}{F_{s}},2\pi \frac{mF_{s}}{M}\right)$, where k∈Z is the time sample index and m∈[0,M−1] is the discrete frequency bin where the number of frequency bins *M* defines the frequency resolution. As a result, the discrete-time STFT is computed as:(14)$$\begin{array}{cc} F_{x}^{h}\left[n,m\right] & =\sum_{k=-\infty }^{+\infty }x\left[k\right]h{\left[n-k\right]}^{*}e^{-j\frac{2\pi km}{M}} \end{array}$$(15)$$\begin{array}{cc} \; & =e^{-j2\pi \frac{nm}{M}}\sum_{k=-\infty }^{+\infty }x\left[k+n\right]h{\left[-k\right]}^{*}e^{-j\frac{2\pi km}{M}} \end{array}$$

For a finite length signal, with a window *h* defined on [−K/2,K/2], the bounds of the summation can be replaced by $K_{min}=\max\left(0,n-K/2\right)$ et $K_{max}=\min\left(M-1,n+K/2-1\right)$. Hence, Equation (15) can be computed by:(16)$$\begin{array}{cc} F_{x}^{h}\left[n,m\right] & =e^{-j2\pi \frac{nm}{M}}\sum_{k=K_{min}}^{K_{max}}x\left[k+n\right]h{\left[-k\right]}^{*}e^{-j\frac{2\pi km}{M}} \end{array}$$(17)$$\begin{array}{cc} \; & =e^{-j2\pi \frac{m}{M}\left(n+K_{min}\right)}\sum_{k=0}^{K_{max}-K_{min}}x\left[n+k+K_{min}\right]h{\left[-\left(k+K_{min}\right)\right]}^{*}e^{-j\frac{2\pi km}{M}} \end{array}$$

As the Gaussian analysis window has an infinite-support length, a threshold Γ is used to obtain a finite number of time samples $K_{m}$ to be considered for the integration at frequency bin *m* such as h[k]≥Γ, $\forall \;k\;\in \;[-K_{m},K_{m}]$. Hence, a small value for Γ leads to a large value for $K_{m}$ and an increase of the computation time. Our implementation based on the Gabor transform uses the discrete-time version of the Gaussian window expressed as: $h\left[n\right]=\frac{F_{s}}{\sqrt{2\pi }L}\;e^{-\frac{n^{2}}{2L^{2}}}$ with $L=TF_{s}$ and with a threshold $\Gamma =10^{-4}$ which provides good results with a reasonable computation time in practice. A Comparison of the different TFRs computed using the proposed methods applied to a real-world instantaneous power electrical signal is presented in Figure 1.

## 3. Non-Intrusive Load Monitoring

### 3.1. Problem Formulation

We address the NILM problem through a supervised machine learning approach illustrated in Figure 2, which requires an annotated dataset of individually recorded HEA for the training step. In our approach, the voltage and current real-valued signals respectively denoted v(t) and i(t) correspond to the input of the proposed method. In a realistic application scenario, an electricity consumption measurement system such as the one we proposed in [18] can be used. The training step (cf. Figure 2a) consists of minimizing the loss function $L(\hat{y},y)$ computed between the predicted label $\hat{y}$ and the ground truth *y*.

The input signals are then used to compute a TF representation from which we derive a set of features used to model each labeled HEA with a unique signature during the training step. Finally, the trained model is used for HEA recognition using the same set of computed input features. The resulting classification method allows the recognition of the closest pre-trained HEA referred to as a “class” according to the machine learning classification terminology.

### 3.2. Electrical Features Computed from Current and Voltage Measurements

In our previous work [15,19], we introduced a set of 90 features, summarized in Figure 3, which are based on the latest IEEE 1459-2010 standard for the definition of single phase physical components under non-sinusoidal conditions [26,27]. Among the 90 proposed features, we selected 34 related to the active and the reactive power (respectively denoted *P* and *Q*) which satisfy the additivity criterion defined in [28]. This property is required for allowing a disaggregation of the mixture signal resulting from several HEAs simultaneously activated before performing the individual HEA recognition process. To this end, an event detection method such as [19] can be used to separate the distinct contribution when several HEAs are switched on simultaneously. In [15], the proposed features are computed for each voltage period and are derived from the Fourier coefficients corresponding to the 15 first harmonics of the voltage frequency. In the present study, we extend the feature definition using the STFT of the discrete-time sampled signals approximated by: $v\left[n\right]=v\left(\frac{n}{F_{s}}\right)$ and $i\left[n\right]=i\left(\frac{n}{F_{s}}\right)$, as detailed below.

#### 3.2.1. Electrical Features Based on Fourier Coefficients

For any frequency index (also called bin) m∈[0,M−1], with M=N (*N* being the finite length of the input signal *x*), the discrete-time Fourier transform of a signal *x* can formulated as:(18)$$\begin{array}{cc} F_{x}\left[m\right] & =\sum_{n=0}^{N-1}x\left[n\right]e^{-j\frac{2\pi nm}{M}} \end{array}$$(19)$$\begin{array}{cc} \; & =\sum_{n=0}^{N-1}x\left[n\right]\left(\cos\left(2\pi n\frac{m}{M}\right)-j\sin\left(2\pi n\frac{m}{M}\right)\right) \end{array}$$(20)$$\begin{array}{cc} \; & =\sum_{n=0}^{N-1}x\left[n\right]\cos\left(2\pi n\frac{m}{M}\right)-j\sum_{n=0}^{N-1}x\left[n\right]\sin\left(2\pi n\frac{m}{M}\right) \end{array}$$(21)$$\begin{array}{cc} \; & =\frac{N}{2}\left(x_{a}\left[m\right]-jx_{b}\left[m\right]\right) \end{array}$$
where $x_{a}\left[m\right]$ and $x_{b}\left[m\right]$ stand for the Fourier coefficients. As this transform is M-periodic, the angular frequency expressed in rad.s${}^{-1}$ associated to a bin *m* is:(22)$$\omega =\left\{\begin{array}{cc} 2\pi \frac{m}{M}F_{s} & \mathrm{if}\;m\in [0,M/2]\\ 2\pi \left(\frac{m-M}{M}\right)F_{s} & \mathrm{if}\;m\in [M/2+1,M-1] \end{array}\right.$$
where m=M/2 is the integer corresponding to the Nyquist frequency, according to the Shannon sampling theorem. In practice, *M* can be artificially increased through zero-padding (i.e., addition of 0 values at the end of signal). This process improves the readability but not the frequency resolution of the output which depends of the analysis window properties [2]. With this in mind, we can express the following quantity, where *P* and *Q* respectively stand for the active power and reactive power:(23)$$\begin{array}{cc} \frac{4}{N^{2}}F_{v}\left[m\right]F_{i}{\left[m\right]}^{*}\; & =\left(v_{a}\left[m\right]-jv_{b}\left[m\right]\right)\left(i_{a}\left[m\right]+ji_{b}\left[m\right]\right) \end{array}$$(24)$$\begin{array}{cc} \; & =v_{a}\left[m\right]i_{a}\left[m\right]+jv_{a}\left[m\right]i_{b}\left[m\right]-jv_{b}\left[m\right]i_{a}\left[m\right]+v_{b}\left[m\right]i_{b}\left[m\right] \end{array}$$(25)$$\begin{array}{cc} \; & =\left(v_{a}\left[m\right]i_{a}\left[m\right]+v_{b}\left[m\right]i_{b}\left[m\right]\right)+j\left(v_{a}\left[m\right]i_{b}\left[m\right]-v_{b}\left[m\right]i_{a}\left[m\right]\right) \end{array}$$(26)$$\begin{array}{cc} \; & =P[m]+jQ[m]. \end{array}$$

#### 3.2.2. New Proposed STFT-Based Electrical Features

The STFT can be viewed as a windowed variant of the Fourier transform, suitable for dealing with non-stationary signals. The main interest of this approach is the capability to convey information about time-varying signal components and the ability to capture transients and impulsive signal components.

Following this idea, we can now replace in Equation (26) the discrete Fourier transforms of the voltage *v* and the current *i* by their discrete STFT as defined by Equation (Equation 16). Omitting the scaling factor $\frac{4}{N^{2}}$, we can introduce the following quantity related to the active power P and the reactive power Q:(27)$$\begin{array}{c} F_{v}^{h}\left[n,m\right]F_{i}^{h}{\left[n,m\right]}^{*}=P^{h}\left[n,m\right]+jQ^{h}\left[n,m\right] \end{array}$$
where $F_{v}^{h}\left[n,m\right]$ and $F_{i}^{h}\left[n,m\right]$ respectively stand for the STFT of the voltage and the current signals using an analysis window *h*. This TF representation can thus be used as the input of a CNN architecture to predict the label of an observed HEA.

### 3.3. Proposed CNN Architectures

The success of CNN is mostly due to their success in image classification tasks and to the development of graphical processing units (GPU), which are of interest for intensive computing. They have successfully been used in a wide variety of signal processing tasks such as speech recognition [29] or music analysis [30], where they are combined with TF representations.

Here, we propose to apply a CNN to the task of HEAs recognition under the assumption that a TF representation conveys a sufficient amount of information valuable in learning useful features. Despite electrical features computed from current and voltage signals frequently being designed and analyzed for HEAs’ identification [9,31,32], we expect that the CNN is able to autonomously learn new features based on relevant patterns present in the data. In our study, we compare two distinct inputs which require dedicated CNN architectures:We compute the TF representation of the instantaneous power signal defined as:
(28)s[n]=v[n]i[n].The spectrogram of this signal looses information about the active and reactive powers. However, it has the advantage of producing a single real-valued matrix that can easily be processed by a classical single input CNN architecture.We compute the product between the voltage TF representation and the complex conjugate of the current TF representation according to Equation (Equation 27) which produces a complex matrix X=P+jQ. The resulting two-dimensional tensor that contains the real and the imaginary parts, can be processed with the proposed CNN architectures. Our first CNN architecture uses a two-channel model and the second one separately process the real and the imaginary part through two distinct CNNs for which their outputs are concatenated at the last layer.

In what follows, we describe the different types of CNN architectures which attempt to exploit all the available information provided by the distinct computed signal representations.

#### 3.3.1. Single-Input CNN Architecture

The CNN architecture depicted in Figure 4 uses as input the TF representation computed from the instantaneous power signal given by Equation (Equation 28). This input layer is connected to four blocks of 32 convolutional neurons with a 3×3 kernel and a REctified Linear Unit (ReLU) activation function followed by a dropout layer and a max-pooling layer. The last hidden convolutional block is connected to two hidden fully connected layers followed by a softmax activation function which produces the final output with the same dimension as the number of expected classes (61 classes considered in this study). Each output node corresponds to the confidence score obtained for each input data. The classification decision rule consists in selecting the HEA class label which obtains the maximal confidence score.

#### 3.3.2. Two-Channel Input CNN Architecture

To bind the information conveyed by the active and reactive powers, the real and imaginary parts of matrix *X* that is computed using Equation (Equation 27) are processed as separated channels into the same CNN model. Both 2D matrices are merged in one tensor of dimension M×N×2 where *P* and *Q* are considered in a depth channel as proposed in [33]. Thus, the proposed architecture in Figure 5 is almost identical to the Single-input CNN architecture in Figure 4, except that each convolution kernel contains an additional dimension, resulting in two channels.

#### 3.3.3. Concatenated CNN Architecture

We propose an alternative CNN architecture depicted in Figure 6. It consists of a combination of two convolutional neural networks for which their final outputs are concatenated. The TF representations of the active and reactive powers are simultaneous processed by two independent single-channel CNNs (depicted in Figure 4). Thus, this model has two branches similar to the single-input CNN architecture. A concatenation layer is used to merge feature maps that are processed by two fully connected hidden layers to produce a single final output.

## 4. Numerical Results

### 4.1. Materials

To make our experiments reproducible, we investigate the publicly available Plug Load Appliance Identification Dataset (PLAID) dataset [17] which contains current and voltage measurements in an American power network at a frequency of 60Hz. This dataset corresponds to long recordings of start-up transients and steady-state operations of 11 different types of HEA: air conditioner, compact fluorescent lamp, fridge, hairdryer, laptop, microwave, washing machine, bulb, vacuum, fan, heater. Each type of HEA is represented by more than ten different instances. For each HEA, three to six measurements are collected for each state transition. The current and voltage measurements of each HEA were post-processed to extract several windows of 800 samples of the current and voltage waveforms such that we have a total of 61 classes of HEA and 16,476 distinct recordings corresponding to steady-state operations or start-up transients.

The resulting TF representations were computed on recordings of 800 samples using a frequency resolution of M=1000 bins. Since the current and voltage signals are real valued, only the positive frequency bins were considered. We then obtain TF representations of size (500×800). Due to memory limitations, we split each TF representation [34,35] into four parts of size (500×200), which are processed as independent inputs and annotated with the same label as the HEA from which the spectrograms are generated. As a result, our experiment considers a total of n=16,476×4=65,904 individuals.

### 4.2. HEA Recognition Results

We address the NILM problem as an electrical load recognition experiment where the reference PLAID dataset is randomly partitioned to obtain the following split: training set (80%) and testing set (20%). Then, we compare the results obtained with the different possible TFRs which are either the spectrogram, the synchrosqueezed STFT or the reassigned spectrogram combined with the different CNN architectures. We also compare these results with our previously proposed approach based on hand-crafted electrical features [15,19]. For each considered experiment, exactly the same training and testing samples were used to comparatively assess the methods. The TFRs were computed using our previously proposed MATLAB implementation included into the ASTRES toolbox [22] with M=1000 and Gaussian window with L∈{60,600}. The neural networks and the machine learning experiments were implemented in Python using Keras and Tensorflow frameworks [36]. During the training step, the number of epochs is set to 40 and an early stopping rule is defined to stop the training when the model accuracy converges (considering 20 epochs for the trigger). The training computation time for each CNN model reaches a maximum of 3 h using a NVIDIA GTX 1080 GPU with 8 GB of RAM.

Table 1 shows the comparative classification results of the proposed CNN models combined with each TF representation. These results are expressed in terms of the classification metrics: F-measure, accuracy, recall and precision that can easily be computed from the resulting confusion matrices [37]. According to Table 1, the overall best results (F-measure of 97.5%) are obtained using the spectrograms of the active and reactive powers combined with the two-channel CNN architecture and are comparable to our method based on P, Q features combined with a Random Forest classification method proposed in [19]. Interestingly, in the single input case based on the TF representation of the instantaneous power, the synchrosqueezing transform obtains the best results (F-measure of 92.1%) and outperforms the spectrogram and the reassignment method. However, when a tensor representation is used for P and Q, the reassignment method outperforms the synchrosqueezing transform despite obtains poorer HEA classification results than those obtained using a classical spectrogram or using our baseline method proposed in [15,19].

More interestingly, we notice that, as expected, the combination of the two signals P and Q lead to the best results since they convey the most amount of information about the HEA signature according to the physical models. However, and despite a better readability of the TF representations provided by the synchrosqueezing and the reassignment methods, these methods do not obtain the best classification results. Thus, one can assume that these post-processing methods, which provide sharpened and sparse representations, have a destructive effect on the relevant information embedded in their TF representations as clearly shown by the HEA recognition scores which are about 10% lower.

### 4.3. Relevance Analysis of the Learned Features

#### 4.3.1. Layer-Wise Relevance Propagation

In order to analyze the HEA patterns learned by the CNN architecture applied on each different TF representation, we use the LRP method [38,39] which allows us to quantify the contributions in the input to compute the classification decision [40,41]. Hence, LRP aims at explaining which input features of a given CNN are the most relevant for obtaining the final classification result. This approach is full of interest, since it is complicated to interpret a CNN model from its learned weight parameters. To this end, LRP proposes to back-propagate the information from the output to the first layer, to obtain a relevance map of the input.

Given a predicted output, LRP proceeds in a reverse manner by iterating over all layers *l* of the network and propagating relevance values from neurons of hidden layers step-by-step towards the input. Given two neurons *i* and *j* connected at two consecutive layers *l* and l+1, each value $R_{i}$ corresponds to the contribution of an input $x_{i}$ that is weighted by $w_{ij}$ when the information is propagated into the neural network for computing to the final prediction. Hence, $z_{ij}=x_{i}w_{ij}$ models the contribution of neuron *i* to make the neuron *j* relevant. This value is then divided by the sum of the contributions at layer *l* connected to neuron *j* to enforce the conservation property [42]. In the present work, we use the LRP-0 method that is deduced from the LRP-ϵ definition expressed as [38]:(29)$$R_{i}^{l}=\sum_{j}\frac{z_{ij}}{\epsilon +\sum_{k}z_{kj}}R_{j}^{l+1}\;\mathrm{with}\;z_{ij}=x_{i}^{l}w_{ij}^{l,l+1}$$
then, the LRP-0 rule is obtained by using ϵ=0 [42]. These rules satisfy the property such as a null input $x_{i}=0$ or the absence of connexion with a weight $w_{i}=0$ leads to a null relevance $R_{i}=0$. However this rule suffers of a numerical instability when the contributions to the activations of a neuron *i* are weak or contradictory. It is to be noted that Equation (Equation 29) is conservative between layers and satisfies the equality $\sum_{i}R_{i}^{l}=f\left(x\right)$ at any layer (l) of the model [43]. In this study, we used the implementation of the algorithm provided by the “iNNvestigate” library [44]. The results are visualized using relevance maps (or heat maps) presented below.

#### 4.3.2. Relevance Maps

We propose to use the LRP-0 method to interpret the discriminatory information contained in the TF representation inputs corresponding to the active and reactive powers used in the two-channel CNN architecture presented in Figure 5. To this end, we present in Figure 7 and Figure 8 spectrograms and the squared modulus of the synchrosqueezed STFT overlayed with the corresponding relevance map computed for two classification examples. Figure 7a,b present the active power P and the reactive power Q spectrograms and Figure 7c,d the squared modulus of synchrosqueezed STFT and the spectrogram of the instantaneous power S with their corresponding overlayed relevance maps respectively computed using the proposed trained 2-channel CNN and the single-input CNN architectures, corresponding to a correctly predicted signal from class 1 (General Electric incandescent light bulb). Figure 8 shows the same TFRs with their corresponding overlayed relevance maps respectively obtained for their trained CNNs, corresponding to a correctly predicted signal from class 32 (a Microwave—Samsung). Interestingly, these TF representations show that, as expected, relevant information is mostly located in the low-frequency range of the signals and mostly embedded around the low-order harmonics. In particular, LRP reveals the area below the first harmonic which contains the most energy located at 120 Hz for *S* or at 60 Hz for *P* and *Q*. The high-frequency area above 700 Hz appears to have a low relevance for each network and lets suppose that high-frequency sampling is of low interest for a HEA recognition application. The computed relevance maps show that there is relevant information between the harmonics of the signal. This is a surprising result in comparison to hand-crafted physics-related features based on the Fourier coefficients of the grid frequency which were previously proposed in [15,19].

To illustrate misclassified examples, we also present in Figure 9, TFRs with their relevance maps for two electrical signals. Figure 9a presents the spectrogram of an individual from class 1 (General Electric incandescent light bulb) which was predicted as class 32 (Samsung microwave) and Figure 9b shows the synchrosqueezed STFT of a signal from class 2 (Frigidaire fridge) predicted as class 16 (Whirlpool washing machine). This latter example clearly present different patterns in their relevance map where the decision was made using the values in high-frequency and between the harmonics parts previously shown as irrelevant. In this case, we believe that the CNN model was trained to use the TFR information located between the harmonics that may be of interest to transient signals (when a load is turned on), however, it could result in to a higher sensitivity in the presence of undesired noise which can be explained by electrical network perturbations. 

## 5. Discussion and Future Works

In this paper, we proposed the first study which investigates the use of concentrated TFRs combined with CNNs applied to a realistic HEA recognition task. We performed a comparison between three different forms of TF representations and evaluated their impact on the CNN classification performance of HEAs, using the publicly available PLAID dataset [17]. First of all, the resulting HEA classification scores showed the validity of our approach with excellent results which outperform existing methods of the literature such as [16] which obtained a F-measure of 77.60% on the PLAID dataset. Moreover, our results are comparable to those obtained with the best approach based on hand-crafted electrical features we recently proposed in [15,19].

A further analysis of the trained CNN-based model confirms our physics-related expectations and shows that the usage of active and reactive powers embedded in a tensor representation outperforms the simple usage the apparent power used as a single-input CNN-based architecture. A surprise in our study is that despite good results, the more readable sharpened TF representations provided by the reassignment and synchrosqueezing methods do not provide the best classification results in comparison to classical spectrograms, except using the single-input CNN-based architecture. This result leaves us with the assumption that relevant information is probably lost in the TF representation during the reassignment (resp. synchrosqueezing) process. This may require a further theoretical study based on classical signal models. Finally, investigations based on LRP are made and show that the trained CNN use the information embedded in the low-frequency harmonics and especially the first fundamental harmonic. This result also confirms the relevance of our hand-crafted electrical features proposed in [15,19] and matches to the resulting set of selected features. Thus, this shows that LRP is a suitable solution for explaining which part of the input of a CNN is relevant for reaching a classification decision [45]. As future work, we expect to investigate more complicated CNN architectures such as Recurrent Neural Networks (RNN) and U-net.

## Figures and Tables

**Figure 1 entropy-22-00911-f001:**
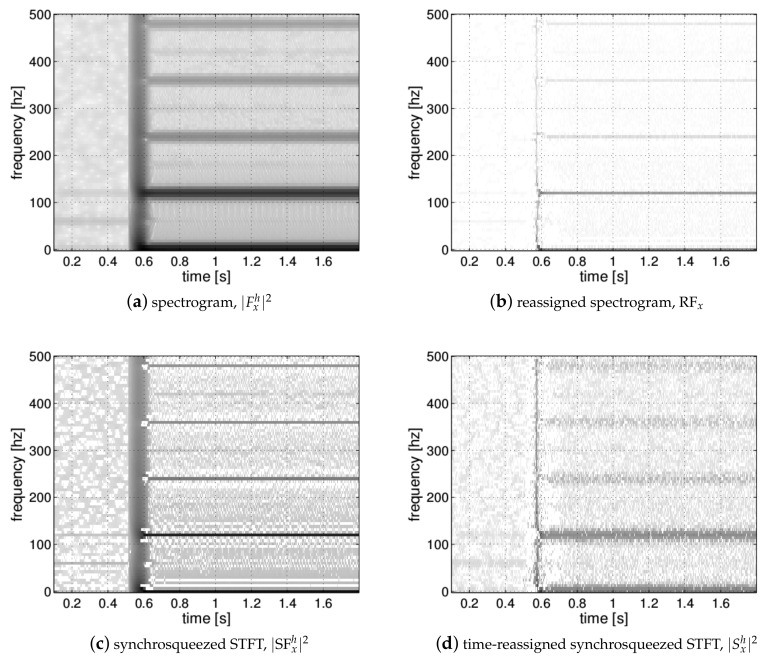
Comparison of the different Time–Frequency Representations (TFRs) respectively provided by the Short-time Fourier Transform (STFT): (**a**) the reassigned spectrogram, (**b**) the synchrosqueezing, (**c**) and the time-reassigned synchrosqueezing (**d**) for the instantaneous power signal of a measured appliance from the PLAID dataset [17].

**Figure 2 entropy-22-00911-f002:**
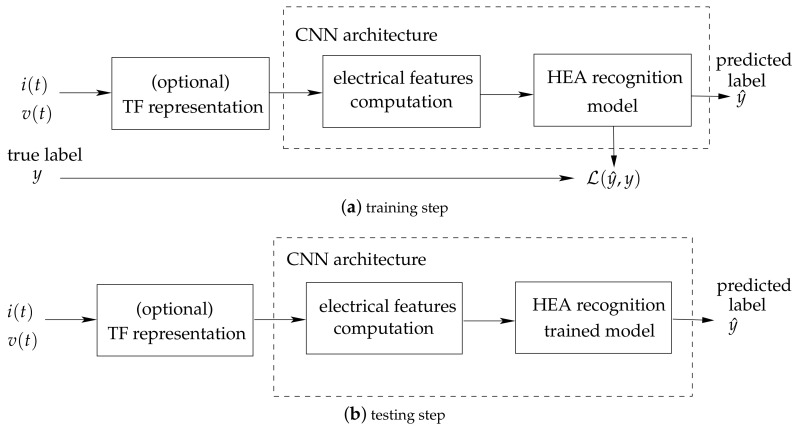
Illustration of the general process for supervised Home Electrical Appliance (HEA) recognition from voltage and current measurements.

**Figure 3 entropy-22-00911-f003:**
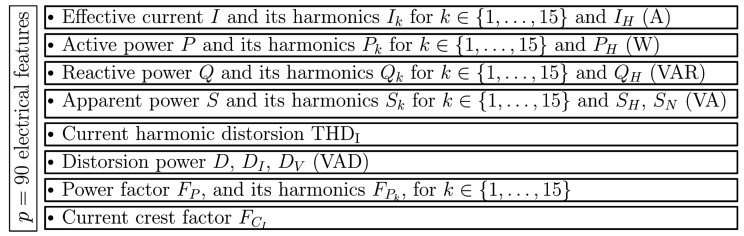
Description of hand-crafted electrical features proposed in [15].

**Figure 4 entropy-22-00911-f004:**
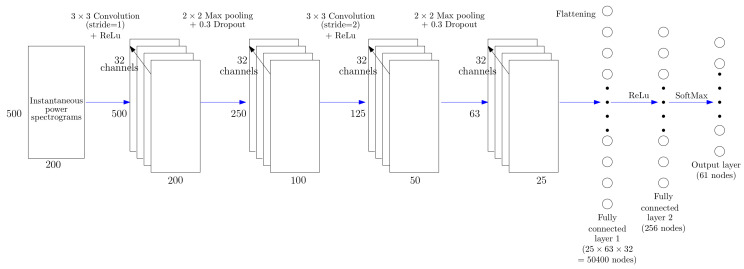
Proposed Convolutional Neural Network (CNN) architecture used for predicting the label of a HEA from the time-frequency representation of its instantaneous power signal.

**Figure 5 entropy-22-00911-f005:**
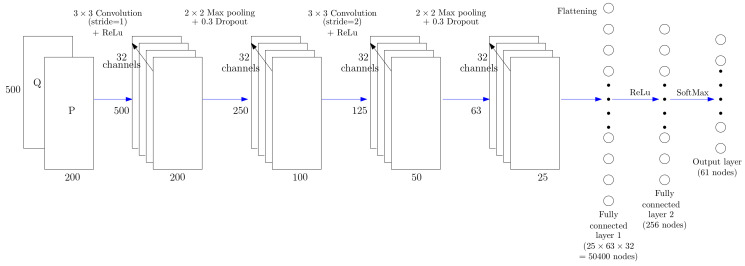
Two-channel input CNN architecture used for predicting the label of a HEA from the time-frequency representations of its active power (P) and reactive power (Q) signals.

**Figure 6 entropy-22-00911-f006:**
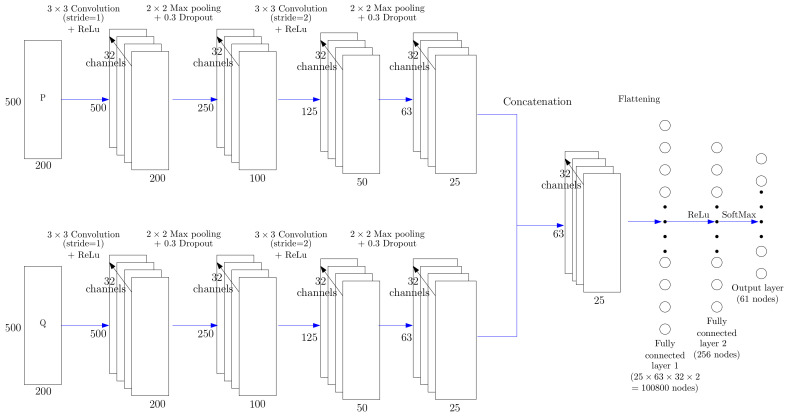
Concatenated CNN architecture used in this research work. The input corresponds to one channel of active power (P) TF representation and one channel of reactive power (Q) TF representation.

**Figure 7 entropy-22-00911-f007:**
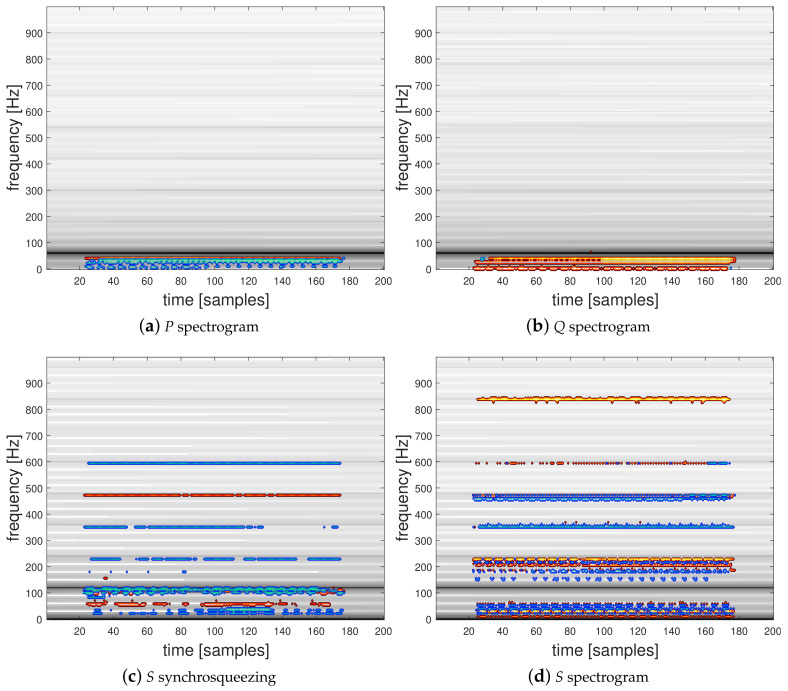
Spectrograms and synchrosqueezed STFT with their corresponding overlayed relevance maps of a correctly predicted individual from class 1 (General Electric incandescent light bulb).

**Figure 8 entropy-22-00911-f008:**
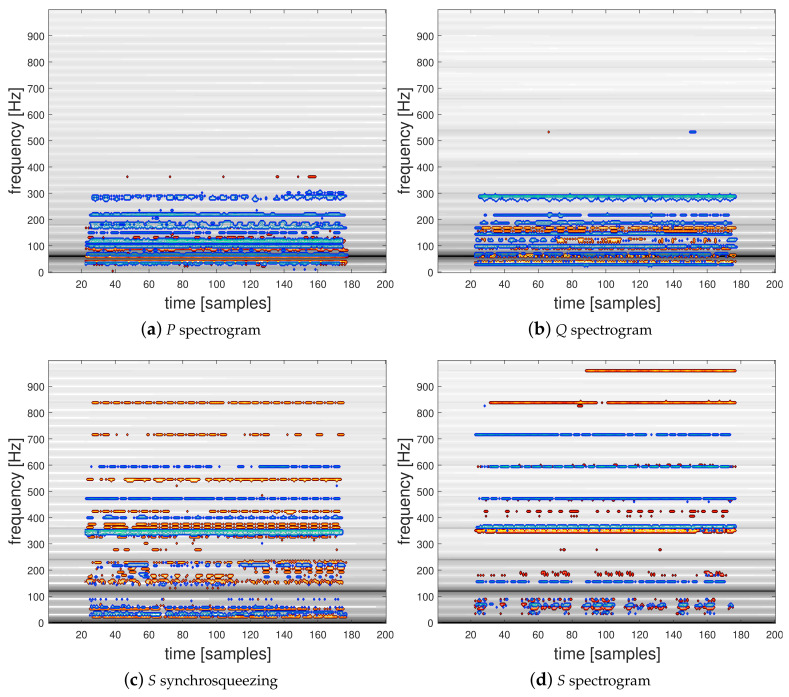
Spectrograms and synchrosqueezed STFT with their corresponding overlayed relevance maps of a correctly predicted individual from class 32 (Samsung microwave).

**Figure 9 entropy-22-00911-f009:**
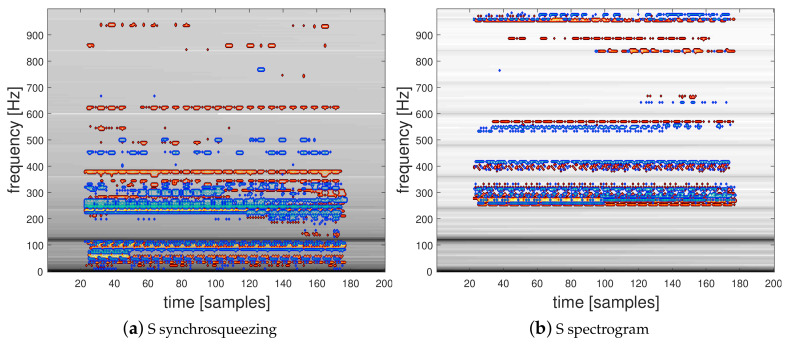
TFRs with overlayed relevance maps of misclassified individuals. (**a**) corresponds to an individual of class 1 (General Electric incandescent light bulb) predicted as an individual of class 32 (Samsung microwave) and (**b**) corresponds to an individual of class 2 (Frigidaire fridge) predicted as an individual of class 16 (Whirlpool washing machine).

**Table 1 entropy-22-00911-t001:** Comparative results (in percentage) of the different HEA recognition methods applied to the PLAID dataset. The window parameter L is empirically chosen to provide the best results.

	Acc	$F_{M}$	Rec	Pre
P, Q + Random Forest [15,19]	97.8	97.7	97.6	97.9
STFT (L = 60, single-input CNN)	87.1	87.2	87.3	88.4
STFT (L = 600, CNN with two channels)	97.7	97.5	97.5	97.9
STFT (L = 600, CNN concatenated)	95.6	95.7	95.5	96.1
Synchrosqueezing (L = 600, single-input CNN)	91.9	92.1	92.4	93.1
Synchrosqueezing (L = 60, CNN with two channels)	85.4	85.0	85.4	86.1
Synchrosqueezing (L = 60, CNN concatenated)	87.2	87.3	87.4	87.9
Time-reassigned synchrosqueezing (L = 60, single-input CNN)	85.8	86.1	86.4	85.9
Time-reassigned synchrosqueezing (L = 60, CNN with two channels)	91.4	91.2	90.9	92.1
Time-reassigned synchrosqueezing (L = 60, CNN concatenated)	92.3	92.3	92.4	91.9
Reassigned spectrogram (L = 600, single-input CNN)	74.4	75.0	74.1	77.3

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
