# Peer review of "On the Use of Concentrated Time–Frequency Representations as Input to a Deep Convolutional Neural Network: Application to Non Intrusive Load Monitoring"

_entropy, 2020, doi:10.3390/e22090911_

Round 1
Reviewer 1 Report
The paper combines time-frequency analysis, known by its capability to efficiently handle non-stationary multi-component signals, and deep learning with CNN, the current state of the art approach for pattern recognition, able to extract relevant signal features. The proposed approach addresses non-intrusive load monitoring (NILM) and consists in identifying a home electrical appliance (HEL) from the energy consumption. The authors investigate the role of TF representation, used as input in a 2D CNN for a pattern recognition task. The experimental results point out the validity of the proposed approach with very good performances, compared with the previous results presented in literature, when other methods were used. The paper includes some interesting ideas on future work in the field making the object of the paper. The paper is interesting and pertinent to the MDPI journal.
The paper is well-written and organized, with some theoretical and practical results. The paper is well documented, including 43 recent references, some of the authors. The paper is easy to follow in its present form, and ideas are presented with clarity. In my opinion, some aspects of the computational complexity of the proposed approach could be of interest for the reader.
Author Response
Dear Reviewer, thank you so much for your comments and for your interest in our work.
We will add a few words about the computation complexity and try to improve the English expression in the revised version of our manuscript.
Reviewer 2 Report
The authors of this manuscript studied to use the concentrated time-frequency representation as input of deep CNN system for non intrusive load monitoring. The present study is a continuation of the authors' previous work on this issue. They proposed a combinatorial approach by the time-frequency analysis and the deep learning algorithm with convolutional neural network(CNN), and demonstrated its effectiveness on the non-intrusive load monitoring of home electrical appliances. I think the results reported in the manuscript are sufficiently novel to satisfy the requirement for publication in Entropy. Therefore, it is suggested to accept the manuscript for publication after the authors make minor revision for the following problems.
- There are serious problems of incomplete information and format confusion in the reference part, which needs to be revised carefully.
- The abbreviation CNN in the title of the paper is recommended to be replaced with the full spelling: convolutional neural network.
Author Response
Thanks a lot for your comments.
We applied the recommended minor changes in the revised version of our manuscript.
Reviewer 3 Report
In this work, the combination of time-frequency analysis and a convolutional neural network (CNN) has been proposed to address non-intrusive load monitoring (NILM) in identifying home electrical appliances (HEA) from its measured energy consumption signal. This study uses time-frequency representation (when synchrosqueezed or not) as the input of the CNN model. The presented model is suitable with respect to the main objectives and supported by the results of the experimented dataset. However, the authors should address the following comments in a revised version before further consideration for publication:
- In the introduction section, provide more information about the HEA recognition task methods from previous studies in the literature.
- Accordingly, the discussion/conclusion part needs improvement. Compare the proposed HEA classification with previous methods in the literature.
- In equation (24), section 3.2.1, the term vb[m]ib[m] has to be positive. When two imaginary numbers (j) are multiplied, the product is negative. Correct the equation (25) accordingly.
- The provided information about “Fourier interpolation” in section 3.2.1 (page 7), needs citation.
- Use the extended terminology "convolutional neural network" instead of CNN, when it appears for the first time in the abstract.
Author Response
Thanks a lot for your very useful comments. We applied the following changes to our paper as follows.
1) In the introduction section, provide more information about the HEA recognition task methods from previous studies in the literature.
response: We have added more citations to previous works related to NILM and HEA recognition. In particular, now we cite a recent method based on CNN which was evaluated on the PLAID dataset.
2) Accordingly, the discussion/conclusion part needs improvement. Compare the proposed HEA classification with previous methods in the literature.
response: we have improved our conclusion and added a reference to a recent previous work based on CNN which obtained poorer results on the PLAID dataset.
3) In equation (24), section 3.2.1, the term vb[m]ib[m] has to be positive. When two imaginary numbers (j) are multiplied, the product is negative. Correct the equation (25) accordingly.
response: You are perfectly right, sorry for this mistake which was fixed. Fortunately, it has no consequence on our method and results.
4) The provided information about “Fourier interpolation” in section 3.2.1 (page 7), needs citation.
response: As proposed, we added a citation and we removed the reference to the expression "Fourier interpolation" that could be confusing.
5) Use the extended terminology "convolutional neural network" instead of CNN, when it appears for the first time in the abstract.
response: We have defined the extended terminology of each abbreviation in the whole paper to ensure that each one is properly defined.